# Lessons from Vaccine-Related Poliovirus in Israel, UK and USA

**DOI:** 10.3390/vaccines10111969

**Published:** 2022-11-20

**Authors:** T. Jacob John, Dhanya Dharmapalan

**Affiliations:** 1Christian Medical College, Vellore 632004, Tamil Nadu, India; 2Department of Pediatrics, Apollo Hospitals, CBD Belapur, Navi, Mumbai 400614, Maharashtra, India

**Keywords:** circulating vaccine-derived poliovirus, inactivated poliovirus vaccine, oral poliovirus vaccine, polio eradication, route of transmission, vaccine-associated paralytic polio

## Abstract

Genetic variants of vaccine poliovirus type 2, imported from an unknown source, were detected in waste waters in Jerusalem, London and New York in early 2022. Wild poliovirus type 2 was globally eradicated in 1999, but vaccine virus type 2 continued for 16 more years; routine use of the vaccine was discontinued in 2016 and reintroduced occasionally on purpose. As an unintended consequence, type 2 vaccine virus variants (circulating vaccine-derived polioviruses, cVDPVs) that mimic wild viruses’ contagiousness and neurovirulence, have been emerging and spreading. To illustrate, in just the past four years (2018–2021), 2296 children developed cVDPV polio in 35 low-income countries. Many assume that virus transmission is via the faecal–oral route. Sustained virus transmission was documented in London and New York, in spite of high standards of sanitation and hygiene. Here, virus transmission cannot be attributed to faecal contamination of food or drinking water (for faecal–oral transmission). Hence, contagious transmission can only be explained by inhalation of droplets/aerosol containing virus shed in pharyngeal fluids (respiratory transmission), as was the classical teaching of polio epidemiology. If transmission efficiency of VDPV is via the respiratory route where hygiene is good, it stands to reason that it is the same case in countries with poor hygiene, since poor hygiene cannot be a barrier against respiratory transmission. By extrapolation, the extreme transmission efficiency of wild polioviruses must also have been due to their ability to exploit respiratory route transmission. These lessons have implications for global polio eradication. It was as a result of assuming faecal–oral transmission that eradication was attempted with live attenuated oral polio vaccine (OPV), ignoring its safety problems and very low efficacy in low-income countries. Inactivated poliovirus vaccine (IPV) is completely safe and highly efficacious in protecting children against polio, with just three routine doses. Protecting all children from polio must be the interim goal of eradication, until poliovirus circulation dies out under sustained immunisation pressure. OPV should be discontinued under cover of immunity induced by IPV to stop the emergence of new lineages of VDPVs, not only type 2, but also types 1 and 3, to expedite the completion of polio eradication.

## 1. Introduction

The Global Polio Eradication Initiative (GPEI) announced on 29 July 2022 the recent detection of type 2 poliovirus (genetically related to Sabin-2 virus) in sewage in Jerusalem, London and New York [1]. One unimmunised person in New York developed polio paralysis in July. In New York State, sewage in Rockland County and New York City were positive for the virus in June; later Orange County sewage became positive and in August, sewage in Sullivan County also had the virus, showing sustained transmission over several weeks [2]. In London vaccine type 2 variant virus was detected repeatedly and continuously from February through July, also showing sustained transmission [3].

Wild poliovirus (WPV) type 2 was eradicated in 1999 [4]. Sabin-2 virus has not been in routine use anywhere since April 2016, after the global switch from trivalent oral polio vaccine (tOPV) to bivalent OPV (bOPV) containing only types 1 and 3 [5]. However, GPEI has stockpile of monovalent type 2 vaccine (mOPV-2) and has used it a few times to control outbreaks due to circulating vaccine-derived poliovirus (cVDPV) type 2, further seeding it in communities with very low type 2 immunity prevalence, contravening the recommendation from experience in Mogilev [6]. The virus importation into Israel, the UK and USA in 2022 was most probably from one such recent reintroduction of mOPV-2.

Against this background, importations of variants of Sabin-2 virus in 2022, attest to the flawed immunisation tactics of the GPEI, particularly in using the vaccine for 16 years beyond its legitimate need, and also reintroducing it even after its global withdrawal.

In recent years, only inactivated poliovirus vaccine (IPV) has been used in the UK (since 2004) and USA (since 1999). News media have suggested that the reason for reintroduction and sustained transmission of poliovirus is the exclusive use of IPV that does not induce intestinal immunity [7]. However, the root cause is the continued use of OPV type 2 when and where it was not needed after the eradication of WPV type 2. We will also present the reason why we do not agree that the lack of intestinal immunity was responsible for the circulation of the vaccine-related virus in London and New York.

The silver lining of the dark clouds of these unfortunate events is that cVDPV-2 in low-income countries, from where it must have been imported, has now received global attention. There, 334 children were paralysed by cVDPV-2 in 2021 and 85 in 2022 (till 18 October) [8]. We hope that course correction will be demanded of the GPEI by global opinion leaders and funders of the eradication efforts, without any more delay.

### 1.1. Genetic Variants of Sabin-2 Virus

The poliovirus genome is single-stranded positive sense RNA, highly prone to mutations. After many centuries of human adaptation, wild poliovirus (WPV) types 1, 2 and 3 have relatively stable genes for structural proteins. WPVs are highly contagious and neurovirulent, the two properties Albert Sabin managed to attenuate through repeatedly selecting and propagating variants with reduced virulence in monkeys [9].

Thus, OPV contains wild-virus-derived polioviruses. During every cycle of multiplication, either in cell culture or human intestines, Sabin original (SO) virus mutates in the direction of the parental WPV genotype. Neurovirulence is regained after very few mutations. OPV is manufactured using SO passaged twice (SO+2) or thrice (SO+3) as seed virus, and next harvest as the final product. As a result, the presence of mutated neurovirulent virus particles in very small amounts in every dose of OPV is inevitable [10,11].

Currently, the entire VP-1 region of poliovirus genome of all field isolates is amplified and compared with that of the virus in OPV, designated ‘Sabin virus’, to distinguish variants drifted further from it. Genetic drift up to 0.5% for type 2 and up to 0.9% for types 1 and 3 are the criteria to classify variants as ‘Sabin-like’ (SL) [12]. The virus in Israel was SL [1]. In London, the earliest virus (8 February 2022) detected in sewage was also SL [3].

Genetic drift of 0.6% and more for type 2 and 1% or more for types 1 or 3 denotes that they are vaccine-derived polioviruses (VDPVs) that have regained neurovirulence and contagiousness. When epidemiological evidence shows community circulation of a VDPV, it is designated cVDPV, which is nearly fully deattenuated, and wild-like. About 1% drift denotes approximately one year of community circulation—as if a ‘molecular clock’ exists.

In Egypt, Sabin-2 virus with genetic drift circulated from 1983 till 1993, causing polio, until stopped with tOPV campaigns—the drift from Sabin-2 virus was about 10% in ten years, validating the molecular clock concept [13]. The genetic distance from SO to WPV is >15%—the two representing polar ends of the spectrum of genetic variation [14].

The shorter drift distance of type 2 indicates it to be less attenuated than Sabin-1 and 3 viruses as far as transmission efficiency is concerned. Sabin-2 virus regains transmission efficiency fast if let loose among susceptible children. For Sabin-1 and 3 viruses, it takes longer; they regain neurovirulence early, but often without transmission efficiency. Globally cVDPV1 and cVDPV-3 emergence has been fewer and far between.

Every dose of tOPV contains very small amounts of neurovirulent mutants [10,11]. They cause vaccine-associated paralytic polio (VAPP), apparently proportional to their numbers, hence very infrequently. Among the three antigenic types, 3 is the commonest cause of VAPP—it is the least attenuated for neurovirulence. Thus, the two attenuation genetics—transmission efficiency and neurovirulence—are dichotomous and regained independently at different frequencies.

### 1.2. Eradication Tactics vis-a-vis Genetic Variants of Sabin-2 Virus

These data had warned polio experts that Sabin-2 had to be very carefully managed as it could emerge as VDPV and cVDPV, if allowed to spread where and when population immunity was not high. High population immunity is the safety wall to block their emergence and spread. The original time frame for eradication was 1988 to 1999, when population immunity against type 2 was induced by both WPV-2 and tOPV [15]. Coverage level of tOPV peaked in 1999, resulting in eradication of WPV-2 in October 1999 [4].

From the vaccinology viewpoint, that was the best time, indeed the only safe time, to withdraw Sabin-2, because of the wall of highest achievable population immunity ever. Such a moment occurs once in history, never to happen again. Sadly, knowledge, courage or wisdom did not prevail; tOPV was continued as business as usual, with the consequences that concern us today.

There was another simpler reason why Sabin-2 had to be withdrawn. When a vaccine (like tOPV), with rare serious adverse reaction (namely VAPP) is used, benefit–risk balance had to be assessed. When WPV-2 was widely prevalent, the benefit of Sabin-2 was greater than its risk of VAPP. As WPV-2 was eradicated, the balance reversed: the risk from Sabin-2 vaccine remained, while benefit disappeared. Continuing Sabin-2 beyond its need, ignoring the negative benefit–risk balance, was against science and common sense.

### 1.3. Iatrogenic Polio vis-a-vis Polio Eradication

Polio caused by the Sabin virus (VAPP) or its variants is iatrogenic. The WHO had issued clear warning and guidance regarding VAPP in the 1980s [16,17]. Countries were asked to monitor VAPP and modify vaccination policy (such as by choosing between tOPV and IPV), according to what frequency of VAPP was acceptable [16,17]. Surprisingly, the GPEI did not follow WHO guidelines: VAPP was not monitored, creating the impression of its absence.

Eradication is precision public health—zero polio and zero poliovirus transmission. The end point was now obfuscated by not monitoring VAPP. Had it been monitored, Sabin-2 VAPP would have alerted countries of its risk without benefit. Should the GPEI have used OPV in the twenty-first century at all, when the eradication target was 2000? The predictable consequence of continuing OPV was VAPP, which went against the spirit of the resolution. When a country (e.g., India) or a WHO region (e.g., South East Asia) or a continent (e.g., Africa) was declared as polio-free, most people accepted it to be true without realizing that unknown numbers of VAPP would have been occurring.

The emergence of VDPVs, cVDPVs, and polio outbreaks due to them, were not exactly predictable but their possibility was known, as we have mentioned in other sections. Obviously the GPEI took a risk and continued with widespread use of tOPV beyond 2000 and the rest is recent history.

### 1.4. Rationale and Consequences of Withdrawal of Sabin-2 in 2016

To withdraw type 2 vaccine virus from tOPV became a necessity. The GPEI waited 16 years to collect ‘evidence’ of polio due to VDPV-2 to be convinced. The probability of emergence of cVDPV lineages was to be expected due to gaps in type 2 immunity in children born since 1999. It was imperative that a wall of high population immunity be created before Sabin-2 withdrawal. The Mogilev experiment (described below) had given us sufficient warning about the silent persistence of Sabin-2 after its withdrawal, unless precaution were robust.

From 2006 to 2015 many lineages of cVDPV were in circulation and all of them had to be interrupted. In 2016, polio was under eradication mode in a globalised world. There was only one way to build a robust wall of population immunity sufficient to interrupt all existing lineages and to pre-empt the emergence of any new lineages. It was to use both IPV and tOPV strategically. IPV had to be in 3 doses and given through EPI, achieving high coverage. Since six-antigen (hexavalent) vaccine including whole-cell pertussis and IPV was available, there was no need for additional injections of IPV. It could have been supplemented by multiple campaigns of tOPV, if the Mogilev experiment was clearly understood. After measuring and ensuring very high population immunity, Sabin-2 could be withdrawn, country by country. These were formidable tasks, but without them Sabin-2 withdrawal in 2016 was too risky.

The GPEI was caught on the horns of a dilemma; cVDPV cases had occurred in 16 countries during 2009 to 2015, but proper preparations were labour-intensive and very expensive. The GPEI took a gamble—with one dose of IPV plus limited tOPV campaigns. Population immunity was not monitored. Sabin-2 was withdrawn from all countries irrespective of population immunity levels.

When fresh cVDPV-2 lineages became visible after a lag time of one year, we knew that cutting corners was most unwise. Even when new lineages emerged in multiple locations, 3 doses of IPV were not introduced. No child anywhere has developed polio after taking 3 doses. Preventing polio in individual children and interruption of cVDPV-2 transmission were both important, but the GPEI neglected the former and tried its best to achieve the latter, so far in vain.

### 1.5. Polio due to Type 2 VDPV in Low-Income Countries

In many countries, cVDPV is currently endemic. In the past four years (2018–2021) outbreaks of cVDPV-2 polio were recorded in 35 countries—25 in Africa and 10 in Eurasia—with polio cases totalling 2296 [8].

In the Democratic Republic of Congo, cases have occurred every year; in Niger, Nigeria, Somalia and Pakistan over three years; in Afghanistan during 2020 and 2021. There were more cVDPV2 polio cases (513) in Afghanistan and Pakistan than of natural polio (354). When polio cases occurred due to importation of cVDPV-2 into Iran, Malaysia, Tajikistan, Ukraine and Yemen during these four years, there was hardly any discourse about it in international media.

Families of children with cVDPV polio would not have realised that the tragedy was due to human error, not nature’s vagaries. No compensation was offered in spite of the root cause of human error. During these four years, cVDPV-1 polio cases were 90 in 8 countries and cVDPV-3 polio cases 7 in one country [8].

### 1.6. History of Polio due to Sabin-2 Variants Goes Back Five Decades

“Those who cannot remember the past are condemned to repeat it”, said George Santayana. There was ample information in the twentieth century about the problem of the Sabin-2 virus mutating and becoming contagious and neurovirulent, if allowed to circulate. None of the successive GPEI leaders was experienced in polio epidemiology in low-income countries, nor well versed in the vaccinology of IPV and OPV. “Superficial knowledge is potentially more dangerous than ignorance. It gives a false sense of security encouraging an ignorant man to persevere in his efforts that can result in huge damage.” [18]. What damage, in polio eradication, could be greater than causing life-long paralysis of 2296 children?

OPV was introduced in the USA in 1961–1962 without assessing safety by clinical trial [19]. Within one year, polio within one incubation period was found in several vaccine recipients—and confirmed as VAPP [20]. VAPP was reconfirmed by the WHO in European countries during the 1970s and 1980s [16,17]. Surprisingly, some unvaccinated children also had polio due to Sabin-2 virus following horizontal transmission from vaccinated children. If there was direct contact with a recently vaccinated child, it was contact VAPP. If no such contact was present, it was community-acquired VAPP. This sinister potential of community spread of Sabin-2 variants was known since 1964. The episode of Sabin-2 variants causing polio during ten consecutive years in Egypt was another reminder of the sinister potential.

### 1.7. Mogilev Experiment

The USSR had been using Sabin tOPV since 1959 in nationwide campaigns, followed by age-based routine immunisation. With abundant forethought, it planned an experiment in the Mogilev District in Byelorussia. After building up high population immunity through two nationwide OPV campaigns (covering 2 months to 10 years once and 7 years next), tOPV was not given in the entire district from March 1963 till March 1966 [6].

The objective was to monitor the duration of immunity and duration of virus circulation in the absence of continued vaccination. Midway, in March 1965, 40 children in six nurseries were given one dose of tOPV. During May to October 1965, poliovirus type 2 was detected in 9 of 392 randomly selected children below 3 years—not from the six nursery cohort, but from others. One of them had facial palsy, which is a well-known form of polio [21].

No type 1 or 2 virus was detected. Antibody studies showed wider prevalence of type 2 and limited prevalence of types 1 and 3 during the lull period. Vaccination with tOPV resumed in April 1966.

This story was reviewed and data and viruses reanalysed in 2003 with clear advice not to reintroduce the type-2 vaccine once it was withdrawn [22]. Yet, having undertaken the world’s largest and most expensive public health programme of polio eradication in 1988, no international consultation was conducted to learn from old hands and from those with different viewpoints.

Sabin-1 or Sabin-3 vaccine could be withdrawn without (or with very little) risk of variants surviving in the community—but Sabin-2 could not be withdrawn at will, as it or its variants may already have been in transmission silently and may have ‘blown up in our faces’ as emerging VDPV and cVDPV. That is exactly what happened.

We already said that it should have been withdrawn immediately after WPV-2 eradication. A wiser course was to begin planning for it in 1993 when the Egypt experience was fresh in memory. That was the ideal time to introduce IPV into EPI, with a view to wean the world off OPV by 1999 [23]. Since that was not done, the next best time was in 2002, three years after the very last case of WPV-2 polio was documented [15]. The next signal was in 2006 when a cVDPV-2 polio caused an outbreak in Nigeria [24].

### 1.8. Route of Infection VDPV-2 in London and New York

One important lesson from the importations of VDPV-2 into London and New York, and its sustained spread locally over several weeks, is about the route of transmission. In London, the Sabin-related type 2 virus was in circulation for more than 6 months (February to July 2022) and in New York it has continued to circulate from March onwards [25,26]. The virus detected in US is genetically linked to viruses detected in London and Jerusalem district, Israel [27]. London and New York have high standards of sanitation and hygiene and contagious transmission by the faecal–oral route is not plausible. The New York State Department of Health has announced that the tap water in New York is completely separate from the sewage system, reinforcing that contamination was not possible [28]. Since the tap water was not contaminated with sewage water, the virus spread was not faecal–oral. The same conclusion applies in London also. The GPEI’s intervention tactics, built on the assumption of faecal–oral transmission of polioviruses, against which intestinal immunity was believed essential, for which OPV was the vaccine of choice, was fundamentally flawed. OPV is given by mouth, but the resultant intestinal infection is not contagious despite viral shedding in stools for several weeks [29,30]. Only infrequently do vaccinated children infect their close contacts and that has never resulted in vaccine virus circulation [31]. Had secondary infection been more frequent, partial coverage of children with OPV should have sufficed for achieving high population immunity. For example, in India 80% coverage was reached in 1990 with 3 doses of tOPV, but 50,000 cases of wild polio were reported (an average 137 paralysed every day) in 1994 [32]. Children had to be reached again and again and given OPV repeatedly, many times, to build up population immunity.

WPV and cVDPV, on the other hand, are contagious. Polio is a disease of infants and pre-school children with those infected having a median age below 15 months, typical of respiratory transmitted infectious diseases such as measles [33,34,35,36].

This stark contrast is explainable as: OPV is adapted to intestinal infection, while WPV is adapted to nasopharyngeal infection as well as intestinal infection. It is during the period of nasopharyngeal virus shedding that WPV is transmitted to others. We are not aware of any research by the GPEI to explore if cVDPV-2 is pharyngeal-adapted. There is no observed or epidemiological evidence for faecal–oral transmission of WPVs [37,38].

The difference between the transmission efficiencies of OPV and cVDPV can be explained by the acquisition of transmissibility by the respiratory route, by deattenuation. Before Sabin promoted OPV, the general teaching was respiratory transmission, for which there is much epidemiological evidence [37,38,39]. Historically, OPV and the notion of the mouth as the natural portal of entry of poliovirus were promoted as a package by Sabin [37,38,39].

Polio was prevalent in all countries irrespective of standards of sanitation and hygiene. Sabin discovered that poliovirus infection occurred in the intestines, resulting in virus shedding in stools. He did not pursue the phenomena of pharyngeal infection, early upper respiratory symptoms of infected children, shedding via droplets/aerosol and the consequent contagiousness. The main argument Sabin put forth was the absence of WPV in the olfactory bulbs of children dying of severe polio to discredit the respiratory route of transmission [40].

In communities highly endemic for WPVs and with very low median age of polio, Sabin OPV is not highly infectious—had it been otherwise, population immunity would have been near-100% by the time tOPV coverage was more than 50%. Infected children shed virus in stools for 4–6 weeks, but despite that vaccine viruses are not contagious. When one million infectious type 1 viruses were fed in tOPV, only 10% of infants in Northern India became infected [41]. In spite of such observations in many low-income countries in the tropical belt of Africa and Asia, clearly negating the belief of faecal–oral transmission, the GPEI seems to have considered the assumption to be dogma that had to be simply believed without argument.

The epidemiology of polio in the pre-vaccine era was typical of respiratory-transmitted diseases. The textbook teaching was also that polio was due to respiratory transmitted infection [42,43]. All exclusively vaccine-preventable diseases, such as polio, measles, rubella, influenza, etc., are respiratory-transmitted—sanitation and hygiene did not stop or reduce their transmission. WPVs had circulated ubiquitously in all high-income and low-income countries, prior to the introduction of IPV and OPV. Even countries known for very high standards of sanitation and hygiene had outbreaks of polio, prevented, controlled or eliminated only by vaccination. Thus, there were ample reasons to conclude that transmission was via the respiratory route, but none for faecal–oral transmission.

The importation of wild virus from Pakistan to Malawi and Mozambique in 2021 and the importation of VDPV-2 into London and New York and its sustained transmission are reminders that the assumption of the faecal–oral route is without any evidence. In both cases, it was probably adults who carried the virus to new territories; if so, reinfected adults were the likely transmission vectors. This will explain the paradoxical decline of WPV polio in Afghanistan and Pakistan in 2021 when all immunisations declined. In 2020 there were 149 cases but in 2021 only 6. When adults wore face masks against the coronavirus epidemic, WPV transmission declined drastically—however, this is only speculation.

### 1.9. The Role of IPV in Polio Eradication

Obviously, Sabin OPV cannot be used in the polio-eradicated world, but IPV can be. If IPV is included in EPI, OPV can be withdrawn in any country that achieves high coverage—90% is reasonable as high coverage. The original World Health Assembly resolution stated: “…eradication efforts should be pursued in ways which strengthen the Expanded Programme on Immunisation…” [44]. IPV as hexavalent vaccine is EPI-friendly, whereas OPV by campaigns is not [29]. Every country that graduates to the withdrawal of OPV will be ‘polio-eliminated’. When all countries eliminate polio, eradication is complete. The polio eradicated world will be using IPV exclusively.

The first step in course correction of the GPEI immunisation tactics is the introduction of IPV, three doses, in all countries that are using OPV. The second step is withdrawal of OPV as stipulated above. Once OPV is removed, no new lineage of VDPV will emerge. As IPV will protect children against polio disease, we can wait for population immunity to interrupt cVDPV transmission. IPV offers better mucosal protection against respiratory transmission than OPV.

Once OPV is globally withdrawn, any poliovirus, WPV, VDPV or Sabin-like, will be a signal of suboptimal population immunity that must be reinforced urgently. Continued environmental surveillance should then remain a crucial strategy to sustain a polio-free world. However, there will be no further need to classify each virus isolate by molecular analysis, thereby saving on expenditure.

This paper is a revised version of a pre-print paper in Qeios [45].

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
