# Peer review of "Lessons from Vaccine-Related Poliovirus in Israel, UK and USA"

_vaccines, 2022, doi:10.3390/vaccines10111969_

Round 1

Reviewer 1 Report

The current global poliomiltys reemergency is a topic of remarkable importance nowdays. Almost when the world seems to reach the erradication of this disease, new and unnespected countries reported cases or enviromental circulations of the viruses type 2, complicating the panorama and moving away from the proposed goal.

The authors present at this time, a little review about some important topics tha coudl help to the cientific comunity to reflext about the reemergency, mainly of Polio virus type 2 at this time in developed country as USA and UK.

In our opinion we ask to the authors to review in more justified and cientific way some aspects listed below. We also invite them to expresss thier ideas taking into account the ethics and impartial mode when they try to valorate which vaccine, in our opinion more than this , which strategy  could be the better alternative to reach the final global polio erradication.

In my opinion, as researchers of everything that is happening, we are call to make a fair and clear assessment of all the scientifics evidences that we have certainly achieved in all those years of investigation and practical development of both vaccines. Nowdays are very well known, and demonstrated the successes and failures, the advantages and disadvantages, the errors and achievements obtained in the processes of production, development and research of both polio vaccines, ignoring them or being biased is not ethical or not helps to the humanity nor to the experts committees to achieve a solution to the problem.

The scientifics evidences that are available at this time speaks for itself, a deep and respectful scientific analysis, with published and impartial valid scientific arguments, will undoubtedly help the entire scientific community that works on this topic. I would dare to say that with polioviruses almost everything was said and proved it. What have we all failed, not only the expert committees, because we are all executors? It has also been more than clear.

I invite to the authors to consult and rewrite impartialy and in more detail the follows issues wherever are mentioned:

1. Producction process of both vaccines, the successes and failures:

2. Advantages and disadvantages of the OPV and IPV vaccines.

3. Definitions of VDPV, cVDPV.

4. Patogeny of Poliovirus,.

5. Environmental surveillance of poliovirus.

6. Circulation of polio 2 in USA y UK.

You can also review:

 Poliovirus Methods and Protocols. (2016). edited by J Martín. New York : Humana Press,

Reviewer 2 Report

The manuscript by T Jacob John and Dhanya Dharmapalan touches on a very important and timely topic of the strategy of global polio eradication initiative (GPEI) that is struggling to achieve its goal for more than 30 years. The main point of the paper is to identify mistakes made by GPEI and to advocate changes. However, the paper is a disappointment for several reasons outlined below. 

First, the style is rather emotional, which is not appropriate for serious scientific discussion. However, the biggest flaw is presenting authors’ personal opinion as a fact without providing references or reasoning supporting the claims. Examples are listed below.

Abstract

Lines 12-13: “Circulating vaccine-derived poliovirus in London and New York, where water supply is safe, attests to respiratory transmission, not faecal-oral.”  This is a speculation, no direct evidence to support this claim, especially in the “Key points” section. 

Lines 19-20. Similarly, the sentence “Virus transmission in high income countries can only be explained by respiratory route, not faecal-oral route, so also in low income countries.” is authors’ opinion. Respiratory transmission is a possible route of transmission, especially in high-income countries, but it does not mean there is not role for fecal-oral transmission, particularly in low-income countries. They cite no study to support this claim, just say that such transmission is “most unlikely” 

Lines 24-25: Claim in the Abstract that IPV provides better protection against respiratory transmission than OPV is not supported by the data, no references are presented, and is not even discussed in the main text.

Lines 34-25: It is unclear in what way Orange County and August Sullivan County were affected.

Line 36: “Sabin virus 2 is no longer in use anywhere”. This is not correct. mOPV2 is being used to control cVDPV2 outbreaks. Close to 1 billion doses were distributed since 2016, seeding even more outbreaks. 

Line 52-53: The meaning of “polar ends of genetic variants” is unclear.

Lines 75-76: “Globally cVDPV1 and cVDPV-3 emergence has been few and far between.”  cVDPV of types 1 and 3 were predominant until the 2016, when switch from tOPV to bOPV occurred. cVDPV2 became widespread only after significant gaps in population immunity to serotype 2 emerged as a result of the ill-conceived switch. 

Line 78: There is no evidence that ‘vaccine-associated paralytic polio’ (VAPP), is proportional to the amount of neurovirulent revertants that are inevitably present in the vaccine. These mutations affect neurovirulence in monkeys, which is not the same as pathogenicity in humans. All cases of VAPP were caused by OPV lots that successfully passed monkey neurovirulence test, and therefore did not contain excessive amounts of mutant viruses. 

Line 91: “As the reverted variants are only about 0.0001% in the vaccine, the risk of VAPP is lower than WPV’s by a factor of 1000 to 10,000.”  Wrong.  OPV3 lots contain ~1% of neurovirulent mutants, OPV1 up to 2-3%, not a fraction of one percent. 

Line 113: “Not monitoring VAPP was another flaw in eradication intervention tactics.”  VAPP was and continues to be actively monitored by GPEI since this phenomenon was discovered. There are numerous papers reporting the findings.

Line 116: “Causing iatrogenic polio was due to error of commission.”  Wrong. Risk-benefit equation was clearly in favor of using OPV, despite a small number of VAPP cases (1 case per 3 million OPV doses). Only after wild poliovirus circulation was stopped the equation changed, which led to the switch from OPV to IPV. It is wrong to call OPV use “risk without benefit”, because OPV was the only tool available for protecting population in low-income countries against polio. 

The thrust of the section starting on line 120 is that OPV2 could have been withdrawn in 1999 right after polio 2 was “eradicated”. This is not only wrong, but also contradicts authors’ statement elsewhere in the paper that OPV withdrawal must be accompanied by the introduction of IPV, not to create immunity gaps. In 1999 the supply of IPV was insufficient for the entire world, and it was insufficient even in 2016, which resulted in the cVDPV2 surge. 

Paragraph starting on line 130: authors are absolutely correct that the right way to switch from OPV to IPV would be to use sequential schedule of three doses of IPV followed by OPV. In reality there was not enough IPV, but despite that the switch we rushed before the conditions were ready, and WHO recommended only one dose of IPV. Only recently SAGE recommended two doses. So it was not that “GPEI took a gamble”, but rather this was a calculated move based on the availability of IPV and the impatient urge to finish eradication. 

Line 181: “OPV was introduced in USA, in 1961-62, without assessing safety by clinical trial.”. This is wrong. Extensive trials were conducted not only in the US, but also in many countries in eastern Europe, Japan, and elsewhere. 

Lines 227-228: see above, the argument that New York has high standards of hygiene and therefore poliovirus cannot transmit by fecal-oral route is very weak. Counties in the state of New York in which polio transmission was detected are populated by families with low income and many children. Therefore, the possibility of fecal transmission is clearly there. 

Line 232: “Only rarely do vaccinated children infect their contacts.” This is not true. Even in the early clinical trials it was demonstrated that vaccine virus can transmit to siblings, caregivers, and other close contacts of vaccine recipients. This was always considered as one of the benefits of OPV by inducing herd immunity. 

Line 237:  The fact that polio attacks children below 15 months of age cannot be used as evidence of respiratory transmission, simply using analogy with measles and other childhood infections. 

Paragraph starting on line 243: It is true that “de-attenuation“ includes both increase of virus neurotropism and transmissibility. But there is no evidence that wild and attenuated strains have different ability to be transmitted by respiratory or intestinal routes. It is well-established that tonsils are the gateway of poliovirus infection and the initial target organ. For a short period after infection poliovirus is shed from tonsils and then colonizes mucosal organs in the intestines, where most of the virus replication takes place. So the respiratory vs. intestinal route of infection is largely a semantic issue. However, the question of where the virus is excreted is not semantic. Most of it is shed in feces, and only very little in tonsils. However what is the relative significance of these routes is still unclear and in fact is not very easy to decide. 

Line 259: Citing low efficacy of OPV in northern India as a proof of “low contagiousness” of attenuated poliovirus is dubious. In this case the low efficacy is most likely a result of tropical enteropathy caused by chronic microbial infections. Thus, I am surprised that authors did not make this argument in support of using IPV in tropical countries. 

Line 270: “Thus there was ample evidence for respiratory transmission but none for faecal-oral transmission.” It would be nice to quote at least one published study supporting this statement. 

In conclusion, this reviewer is sympathetic to authors’ insistence on the revision of the GPEI strategy. There is clearly a role (perhaps even the leading role) for IPV in future immunization policies. However, the paper is very biased and contains many unsubstantiated claims. The discovery of silent circulation in high-income IPV-only countries that is used as a springboard for their analysis clearly shows that IPV does not (and probably cannot) prevent virus circulation despite being highly effective in protecting individuals from paralysis. Modern societies have an increasing number of people who do not have immunity, either because of their refusal to vaccinate, or because of medical reasons, e.g. immunosuppressive therapy. Therefore, the immunization policy must aim not only at protecting individuals, but the entire community by preventing even silent virus circulation, which IPV cannot accomplish. For these reasons the manuscript may need to be seriously revised not to mislead the readers. 
